# ARACAM: A RGB-D Multi-View Photogrammetry System for Lower Limb 3D Reconstruction Applications

**DOI:** 10.3390/s22072443

**Published:** 2022-03-22

**Authors:** Marco A. Barreto, Jorge Perez-Gonzalez, Hugh M. Herr, Joel C. Huegel

**Affiliations:** 1Tecnologico de Monterrey, Escuela de Ingenieria y Ciencias, Av. General Ramon Corona 2514, Zapopan 45138, Mexico; marco.barreto@exatec.tec.mx (M.A.B.); jhuegel@tec.mx (J.C.H.); 2Unidad Académica del Instituto de Investigaciones en Matemáticas Aplicadas y en Sistemas, Universidad Nacional Autónoma de México, Parque Científico Tecnológico de Yucatán, Km 5.5 Carretera Sierra Papacal-Chuburna, Mérida 97302, Mexico; 3Lisa K Yang Center for Bionics, Massachusetts Institute of Technology, Cambridge, MA 02142-1308, USA; hherr@media.mit.edu

**Keywords:** point cloud, iterative closest point, dynamic reconstruction, multi-view

## Abstract

In the world, there is a growing need for lower limb prostheses due to a rising number of amputations caused primarily, by diabetic foot. Researchers enable functional and comfortable prostheses through prosthetic design by integrating new technologies applied to the traditional handcrafted method for prosthesis fabrication that is still current. That is why computer vision shows to be a promising tool for the integration of 3D reconstruction that may be useful for prosthetic design. This work has the objective to design, prototype, and test a functional system to scan plaster cast molds, which may serve as a platform for future technologies for lower limb reconstruction applications. The image capture system comprises 5 stereoscopic color and depth cameras, each with 4 DOF mountings on an enveloping frame, as well as algorithms for calibration, segmentation, registration, and surface reconstruction. The segmentation metrics of dice coefficient and Hausdorff distance (HD) show strong visual similarity with an average similarity of 87% and average error of 6.40 mm, respectively. Moving forward, the system was tested on a known 3D printed model obtained from a computer tomography scan to which comparison results via HD show an average error of ≤1.93 mm thereby making the system competitive against the systems reviewed from the state-of-the-art.

## 1. Introduction

The most common cases that cause lower limb amputations are diabetes, vascular problems, neuropathy and trauma. Over 150 thousand amputations take place each year in the United States alone from these causes [1]. As an example of a lower to middle income country (LMIC), Mexico is of particular interest due to its high levels of both diabetes and lower-limb amputations. According to the Mexican Institute of Social Security, in 2014 alone, there were nearly 900 thousand amputees in the country, to which are added 75 amputations per day. This means that each year Mexico is accumulating more than 70 thousand people with lower-limb amputations, forming now part of those people with disabilities [2]. This generates economic problems and dependency for their caregivers, their families, and the community. This is why prostheses have an important role in recovering a percentage of mobility in amputees, providing autonomy to the person in carrying out activities, as well as complementing the person in a social and cultural aspect, preserving their individual identity [2].

The most important part of the prosthesis is the socket, because it is uniquely shaped for each individual. It forms part of the four sections in a lower limb prosthesis, i.e., the socket, knee, pylon, and foot [3]. While several prosthetic parts can be obtained in large quantities, materials, and sizes. The socket is the component that interfaces directly with the amputee’s residuum and where the load transfer occurs. Thus, a correct fit must be achieved. A favorable socket design stimulates muscle growth and relieves pressure in sensitive areas providing comfort to the user [4]. On the other hand, improper fitting may result in skin problems and tissue injury caused by unbalanced weight and friction [5].

That said, the socket requires a manufacturing process that ensures a match with the user’s residuum measurements and shape. Currently, design and manufacture continue to be handcrafted since all the fabrications are unique and unrepeatable [5]. The overall socket fabrication can be seen in Figure 1. First, a prosthetist performs an evaluation by taking limb measurements Figure 1a and a cast mold Figure 1b is taken. Subsequently, the mold is filled to obtain a model that describes the geometry of the residual limb. Afterward, post-processing is necessary to rectify, add and remove material, to ensure that the dimensions match with those obtained Figure 1c, but also to relieve pressure in sensitive areas while loading more resistant regions. Then, the socket is manufactured by laminating the mold and then curing it. Polypropylene shown in Figure 1d is often a material of choice for check sockets. Finally the socket gets polished, as shown in Figure 1e. The time and number of iterations invested in the process depend on the experience of the prosthetist, since obtaining a fitting socket consists of trial and error. Typically the participant will receive a check socket, return for adjustments to get a provisional socket that will last from 6 months to a year before the residuum strengthens and changes in volume. Here again, a check socket must be fabricated, and afterward, a definitive socket is delivered and can be used for several years before wear and morphological changes require the process to start all over again [6].

Computer vision appears to be a tool that benefits these processes through digital reconstruction with the use of photogrammetry to obtain a digital surface model. This technique is used to obtain detailed information from photographs. Then, algorithms take these images and estimate their position to generate a 3D point cloud. This can be done with multi-view stereo cameras as mentioned in [7]. In the literature, the study of lower limb reconstruction has advanced Computer-Aided Design (CAD), which assists in socket manufacturing. Digital models allow iteration at a minimal cost and reduce the time taken for product manufacture. Moreover, digital models allow accurate geometry and additional relevant information to be gathered for socket fabrication. Useful existing technologies and methods involve not only photogrammetry, but also 3D scanning with structured light, 3D laser scanning, computed tomography (CT), and magnetic resonance imaging (MRI). Image acquisition systems such as CT and MRI can provide information about the internal and external tissue of the residuum, information that can be useful in the design of the prosthesis [5]. These methods are, however, more expensive compared to a photogrammetry based system, they are not portable, acquisitions can only be made with the participant lying down, the images can have various artifacts due to the long acquisition periods and in the case of CT equipment, there is ionizing radiation, which adds exposure when making multiple acquisitions in a short period of time [4,6,8]. In the case of structured light-based camera arrays, interference can be generated between the light patterns of the multiple cameras, which can affect the estimated depth information. Therefore, it is necessary to multiplex the cameras during the acquisition, which can generate artifacts in the acquisition of moving objects [9,10]. Finally, a hand-held 3D laser scanner only performs a partial capture of the lower residuum, therefore an array of laser scanners is necessary to capture the entire target in real-time or if any motion is desired, resulting in the technique being expensive [11]. For these reasons, the present work focuses on designing a lower-limb residuum capture system based on multi-view photogrammetry. This is a low-cost and portable solution, capable of easy manipulation and of making captures simultaneously.

### State of the Art

Multi-view systems are based on photogrammetry, which assumes to have multiple repeated elements of the same object in multiple images. This is so as to obtain information about physical objects and the environment by capturing a sequence and processing the sequence to extract data such as properties and measurements as mentioned by Wang and Riel [7,12]. Most of the algorithms employed deal with the topic of 3D reconstruction. Then, a triangulation of points can be made from the images and the world. So a point cloud can be made from this point triangulation. This can be approached via a single camera that is re-located to have different views of a target or via an array of cameras pointed at the whole target object while having overlapping areas of said object [7].

Several studies on lower limb reconstruction using multi-view photogrammetry have taken place. Taqriban et al. conducted a study on close-range photogrammetry by using a digital reflex camera and Autodesk^®^ Recap Photo software to obtain a 3D model. The lower residual limb of a participant was scanned 360° with two different shooting angles, a 3D model of the external surface was obtained with Recap and then a manual rectification was performed. It is worth mentioning that the manipulation process towards the socket manufacture is still digitally handcrafted with this method [13]. Wu et al. proposed a system with six RGB-D cameras in an array that identifies landmarks and measurements taken from a foot. The results showed a 3D capture system able to scan a foot, obtain multiple point clouds (PCs), and merge them using an iterative closest point (ICP) algorithm. In addition, twelve anatomical landmarks of the foot can be detected automatically without manual intervention [14]. Solav et al. present a multi-camera array to analyze lower body muscular deformation through Digital Image Correlation (DIC) by using 12 RGB cameras. The methodology proposed includes a stage of distortion correction, stereo calibration using a cylinder, and 2D-DIC of stereo image pairs. For the final 3D reconstruction, the fusion of adjacent surfaces is carried out by means of Delaunay triangulation. The proposed system was tested with a rigid cylinder and with the lower leg while the subject performed ankle plantarflexion [15]. Cullen et al. have performed the evaluation of the quality for 3D reconstruction of transtibial sockets using a smartphone in the combination with the commercial Autodesk^®^ ReCap Photo software and a genetic algorithm for the optimal selection of the photos used in the reconstruction [16]. Mohd Sobani et al. report a reconstruction of a representative model of a transtibial socket from a sequence of uncalibrated 2D images. The proposed system is based on a rotating base and a digital camera. For 3D reconstruction, they integrate Canny’s contour detector, the Random Transform algorithm, and the Delaunay triangulation to map the connected points on a 3D surface [17]. Ballit et al. merge the information of 4 RGB-D Kinect v2 cameras. The methodology they propose consists of a stage of generation of point clouds through depth information, followed by error compensation using a chessboard, outliers elimination using statistical outlier removal algorithm, and iterative alignment using the method based on the Singular Value Decomposition (SVD). The validation of the methodology proposed by these authors was performed by means of a known cylindrical object [18]. In prior work, we presented a semi-automatic system capable of capturing a whole target object with four fixed RGB-D cameras that surround the target. Thus, the system can minimize the involuntary movements of the body during capture by taking captures almost simultaneously. A human hand was used to demonstrate its capabilities for non-static and deformable targets [9].

All of these studies have important attributes that make strong research in their field, the use of different approaches or a combination of these can be useful for this work to move forward the lower-limb residuum prosthesis socket design and fabrication. Some of the research described, however, use commercial software to rebuild; were tested and validated with cylindrical models considering only the circumference in the reconstruction; they do not include the design of a specific capture system for applications of subjects with lower limb amputation, or they use a single camera, thereby making it difficult to capture–at one time–the entire object of interest.

The proposed work is directed in the design and construction of a novel yet specific system for the acquisition and external reconstruction 3D of the residuum for participants with transtibial amputation. The array-of-cameras capture system (ARACAM), integrates 5 RGB-D cameras distributed optimally to capture the object of interest appropriately. Multi-view information is automatically integrated using a photogrammetry-based approach. The implemented algorithm integrates a stage of calibration, segmentation, generation of point clouds, alignment of multiple views, and a filtering stage. Finally, the proposed capture system is validated by rebuilding plaster molds, as well as a final test with a participant with transtibial amputation. The ARACAM project is available at https://github.com/marcoagbarreto/ARACAM (accessed on 1 March 2022).

The following sections describe in detail the proposed methodology for the design of the capture system and its validation, followed by the results found, discussion, and conclusion of the work developed.

## 2. Materials and Methods

This section presents in detail the different stages carried out for the development of the system (Figure 2). The methodology begins with the mechanical design of the array-of-cameras capture system (ARACAM); followed by calibration of the cameras using only a checkerboard; the development of an automatic segmentation algorithm, the creation and alignment of 3D point clouds, and finally, the validation of the results by using Dice Coefficient (DC) and Hausdorff Distance (HD) metrics.

### 2.1. Capture System: ARACAM

The term ARACAM is an abbreviation that stands for an array of cameras and it is meant to describe the mechanical design and physical implementation. This section deepens into the optimal distances between the camera and the target, leading to the mechanical design.

#### 2.1.1. Resolution

The ARACAM uses 5 Intel^®^ RealSense^®^ D435i RGB-D camera. This component meets the pre-determined design requirements, by capturing depth and color images and an accuracy of ≤2% at 2 m. Additionally, the camera uses stereovision that allows multi-view setups.

The optimal distance from the camera to target gets calculated and the resolution that ensures a resolution less than 0.5 $\frac{\mathrm{mm}}{\mathrm{px}}$ considering the range distance suggested by the manufacturer. The distance from the cameras to the objective, and the resolution, share a common relationship through the focal distance, as shown in Equation (Equation 1), which was obtained from prior work [9],
(1)$$\begin{array}{c} f=\frac{B}{G}\left(l\right) \end{array}$$
where *f* is the focal distance, *B* is the image size, *G* is the target size and *l* is the distance to the target.

From Equation (Equation 1), *B* holds the information of the image width *i* and height *j*, while *G* is the size in mm of the objective for width and height denoted by *w* and *h*. Then, the resolution of the image can be calculated with Equation (Equation 2), obtained from previous work [9],
(2)$$\begin{array}{c} res=\frac{G}{B}=\frac{\left(i,j\right)}{\left(x,y\right)} \end{array}$$
where *w* and *h* are the real width and height in mm captured by the camera, *i* and *j* is the number of pixels regarding the camera’s width and height, res is the resulting resolution in terms of $\frac{\mathrm{mm}}{\mathrm{px}}$. Since pixels are squared, the resolution can be calculated with only one axis component.

The camera’s manufacturer establishes that the ideal usage range is from 200 mm to 3 m for a proper depth estimation for the D435i, but may go up to 10 m. Given these specifications, The dimensions of the frame is proposed to have a diameter of 610 mm. Next, the distance from the cameras to the objective given these parameters gets calculated. An experimental measurement is taken from the circumference *C* of a healthy man’s leg, with a 1.85 m height and a body mass of 73 kg to find a radius *r* in Equation (Equation 6):(3)$$\begin{array}{cc} c & =470\;\mathrm{mm} \end{array}$$(4)$$\begin{array}{cc} c & =\pi (d) \end{array}$$(5)$$\begin{array}{cc} d=\frac{470\;\mathrm{mm}}{\pi } & =149.6\;\mathrm{mm} \end{array}$$(6)$$\begin{array}{cc} r=\frac{d}{2}=74.8\;\mathrm{mm} & \approx 75\;\mathrm{mm} \end{array}$$
where *c* is the circumference, *d* is the diameter, *r* is the radius.

Then in Equation (Equation 7) the distance *l* from the frame circumference to the target is obtained:(7)$$\begin{array}{c} l=305\;\mathrm{mm}-75\;\mathrm{mm}=230\;\mathrm{mm} \end{array}$$
the resulting distance, l=230mm, lies within the ideal usage range with an additional 30mm gap to avoid the 200 mm lower limit. In this setup, the D435i uses a color resolution of 1280×720 px, with a field of view of 69.4×42.5, horizontal and vertical respectively, as provided by the manufacturer [19]. At this distance, the horizontal range is found with Equation (Equation 10), similarly, the height range is obtained in Equation (Equation 10). The camera is placed in a vertical stance to cover the most length possible of the target. Then, the resolution is calculated with Equation (Equation 12):(8)$$\begin{array}{cc} \frac{w}{2}=\tan\left(34.7^{\circ }\right)\left(230\;\mathrm{mm}\right) & =159.26\;\mathrm{mm} \end{array}$$(9)$$\begin{array}{cc} w & \approx 318.52\;\mathrm{mm} \end{array}$$
(10)$$\begin{array}{cc} \frac{h}{2}=\tan\left(21.25^{\circ }\right)\left(230\;\mathrm{mm}\right) & =89.44\;\mathrm{mm} \end{array}$$
(11)$$\begin{array}{cc} h & \approx 178.88\;\mathrm{mm} \end{array}$$
(12)$$\begin{array}{cc} res=\frac{318.52\;\mathrm{mm}}{1280\;\mathrm{px}} & =0.24\frac{\;\mathrm{mm}}{\;\mathrm{px}} \end{array}$$
where *w* and *h* are the width and height length of the field of view.

When finding a maximum resolution of 0.24 $\frac{\mathrm{mm}}{\mathrm{px}}$ for this setup at a distance of 230 mm, it is found that lower values increase the resolution. The resolution obtained of 0.24 $\frac{\mathrm{mm}}{\mathrm{px}}$ is less than the 0.5 $\frac{\mathrm{mm}}{\mathrm{px}}$ specification, meaning a better image resolution for the application (see Figure 3). The cameras may be adjusted on the rails to increase or decrease the distance to the target from a minimum of 230 mm up to 400 mm, which allows flexibility for use with larger objects.

In the case of 3D video reconstruction. The cameras allow simultaneous capture and streaming thus supporting up to 90 frames per second (fps). The faster the frame rate, the smoother the transitions will be between reconstructions, thereby effectively achieving an end result of 3D video data is possible. Higher fps facilitate deformation analysis by determining the changes from one frame to another.

#### 2.1.2. Mechanical Design

The frame is designed and manufactured to support and hold the cameras in place, it is made out of laser-cut and bent aluminum sheets capable to support the system components. Figure 4a shows a top view of the frame, where cameras are spaced every 90°, thereby equidistant when using 4 cameras and the calculated radius obtained in Equation (Equation 6) to the target. The number of cameras and their radial position was obtained through trial and error. Since plaster cast molds and residuums have convex shapes, our results show no occlusion, thus the choice of 4 cameras at 90°. These locations are marked in blue dots. Furthermore, Figure 4a displays the overlapping angles between cameras. The structure has an overall dimension of 534.50 mm height and an inner diameter of 610 mm where the camera lenses are placed, as shown in Figure 4b. The figure also shows the adjustable locator with the ball head mount for all 4 cameras and the one end camera that captures the end view of the residuum. Previous work utilizes only 4 radial cameras, demonstrating a robust full capture of the target [9]. This study reveals the need for a fifth camera directly pointing at the head of the residuum, as it contains important skin information post-surgery [8]. So, with a total of 5 cameras, 4 radially placed every 90° and 1 camera at the base capturing the end view of the residuum successful captures are achieved. This represents the mechanical design.

The structure comes into close proximity with the amputee when scanning. This is why, the system works in a non-contact scheme, where the design in *C* and the system can be accessed from a wheelchair or other sitting position. This prevents accidents or captures data corruption caused by bumps or scrapes. Figure 5 displays the example scenario of this interaction, highlighting three main parts, the roller chair with the amputee, the fixed frame, and the cameras.

For the placement procedure, the participant is seated at the edge of a raised surface, in this case, a roller chair. The residuum rests flying in a state of relaxation. Then the roller chair is slid towards the fixed structure by the system operator. While gently moving the participant and chair, the residual limb is centered within the camera angles. The opening in *C* allows the system to move comfortably to enclose the amputation, keeping it in the center of the structure. The residuum is kept at rest during the scan process. In the end, the operator removes the roller chair from the structure in the same way.

### 2.2. Calibration

Calibration is used to obtain intrinsic and extrinsic system parameters. Intrinsic parameters describe the position of the internal components, in this case, the distance from one color sensor to the other and the position of the projector, depth scaling factor, and the distortion parameters. These are needed to map out RGB-D images into PCs. While extrinsic parameters describe the rigid transformations of a global coordinate system, in other words, where are the cameras located in the physical space. The transformations are useful for global registration, helping as a first rough alignment between PCs.

Since RealSense cameras are used, most of our implementations rely on base code provided by the manufacturer from the official repository [20]. Intrinsic parameters are given by the manufacturer and may be accessed through the camera’s firmware. While extrinsic parameters must be calculated. Intel^®^ already provides a calibration demo for the D435i cameras [20]. All cameras must be facing inwards, viewing a common 6×9 checkerboard as the calibration item as shown in Figure 6. The calibration is performed by using a Kabsch algorithm [21]; this algorithm finds the optimal rotation matrix between two sets of points. It superimposes both sets of points and minimizes the Root Mean Square Deviation (RMSD). This is realized in three stages.

First, the centroids of two point-sets P and Q are calculated (Equation 13). Each is the average position of the overall points, and a translation of the centroid to the origin of the coordinate system is performed (Equation 14). This is obtained by subtracting the centroid from all of the point coordinates: (13)$$\begin{array}{cc} C(x,y,z) & =\frac{\sum_{k=1}^{n}x_{k}}{n},\frac{\sum_{k=1}^{n}y_{k}}{n},\frac{\sum_{k=1}^{n}z_{k}}{n} \end{array}$$(14)$$\begin{array}{cc} t(x,y,z) & =(x-x_{c},y-y_{c},z-z_{c}) \end{array}$$
where C is the resulting centroid, t is the translation, *x*, *y* and *z* are point coordinates in a 3D space. Second, the covariance matrix H is computed in Equation (Equation 15):(15)$$\begin{array}{c} H=P^{T}Q \end{array}$$
where P is the reference matrix and Q is the matrix to be rotated. Third, a Singular Value Decomposition (SVD), is computed from the covariance matrix H in Equation (Equation 16). Then, a check-sign takes place to ensure a proper rotation in Equation (Equation 17). Finally, the rotation matrix R is calculated in Equation (Equation 18). This returns a superimposed matrix that "fits" the reference matrix by minimizing the RMSD in Equation (Equation 19): (16)$$\begin{array}{cc} H_{n\times m} & =V_{n\times n}S_{n\times m}W_{m\times m}^{T} \end{array}$$(17)$$\begin{array}{cc} d & =\mathrm{sign}\{\det\left({\mathrm{WV}}^{T}\right)\} \end{array}$$(18)$$\begin{array}{cc} R & =W\left(\begin{array}{ccc} 1 & 0 & 0\\ 0 & 1 & 0\\ 0 & 0 & d \end{array}\right)V^{T} \end{array}$$(19)$$\begin{array}{cc} \mathrm{RMSD}(p,q) & =\sqrt{\frac{1}{n}\sum_{k=1}^{n}{\left|p_{k}-q_{k}\right|}^{2}} \end{array}$$
where H is an n×m covariance matrix, V and W are orthogonal, and S is diagonal. *p* and *q* are the *k*-th points of matrices P and Q respectively.

Once the calibration is complete, the program saves the intrinsic and extrinsic parameters. Figure 6 displays the view from all 5 cameras looking at a common checkerboard target which is serving as the calibration object. Additionally, all the images display a green bounding box corresponding to the book underneath the checkerboard, correct orientation and dimensions validate a successful calibration of the cameras.

### 2.3. Segmentation

By default, the cameras take raster RGB and depth images of the whole scene. This means that not only is the target object present but also background and spurious surfaces add noise to the reconstruction. This is why image segmentation is needed. Segmentation allows keeping only the desired object by removing unnecessary elements. OpenCV [22] and Scikit-image [23] libraries are used for their segmentation algorithms. First, the original RGB image is taken. Then the background gets removed by using a depth threshold of 500 mm, which leaves the target object and the table previously covered with a green sheet. Then, a color segmentation is employed by setting a color range filter *U* with an upper and lower bound $U=x_{k}-x_{1}$, the Hue, Saturation, and Value (HSV) color space is used to control the selection of darker and brighter values of the same color. This extracts the pixels corresponding to any green color, within the selected range, on the image and thus belonging to the sheet. This operation creates a binary image that shows the pixels corresponding to the color range previously selected.

Consequently, an algorithm checks an area with the biggest connected elements in the image, assuming the desired object is the largest on the remaining scene. To do so robustly, Otsu’s Thresholding algorithm [24] computes and finds the intensity threshold value that separates pixels into two classes determined by maximizing its variance in Equation (Equation 20):(20)$$\begin{array}{c} \sigma_{\omega }^{2}\left(th\right)=\omega_{0}\left(th\right)\sigma_{0}^{2}\left(th\right)+\omega_{1}\left(th\right)\sigma_{1}^{2}\left(th\right) \end{array}$$
where ω are the probabilities of the classes, th is the threshold, and σ are the variances. The resulting threshold is used to binarize the image.

Next, connected pixels of the same value represents an area. So, for every region, pixels are counted, the region with the highest amount of pixels is assumed to be the target object.

Afterward, median and erosion filters are applied to the binary image from the resulting largest element in the image. These filters are used to smooth out the edges of the remaining region, The median filter selects the median value pixel from a 15×15 squared mask. This process is repeated 10 times swiping the whole image with the mask represented in Equation (Equation 21):(21)$$\begin{array}{c} median\left(k\right)=\frac{k_{\left(n+1\right)}}{2} \end{array}$$
where median is the resulting filter, *k* are the values inside the mask. This filter is useful to smooth the edges from the previous colored segmentation.

The erosion filter is a morphological operator, it computes the local minimum within a 3×3 mask. This erosion takes place on the binary image to slim the target and iterates 10 times. This helps to remove possible outliers that neither the depth threshold nor color segmentation could not resolve. The erosion of a binary image can be described in Equation (Equation 22):(22)$$\begin{array}{c} A\Theta B=\left\{\begin{array}{c} min\;A(u)\\ u\in B⋂A \end{array}\right\} \end{array}$$
where A is the matrix image and B is the mask.

At last, the target object segmentation is obtained by performing a bit-wise operation of the binary image over the original image. While the choice of techniques for segmentation appears somewhat arbitrary, they present a robust, yet simple, solution for the given application.

After the segmentation process completes, the 5 RGB and 5 depth images must be aligned before converting them into a single Point Cloud (PC). To do so, epipolar geometry is needed to find a correspondence between the depth image and the color image. This is also the case for stereovision with the D435i cameras, for estimating depth. From the factory, the D435i has the parameters needed to align color and depth images. Following, a program converts aligned RGB-D images into a PC by mapping the depth information using the intrinsic parameters from each camera. This is performed by using Shreeyak’s piece of code as a base and modifying it to meet the needs of this work [25]. When the color image is segmented, only valid color data is processed with the depth information, meaning that everything out of the segmented region is dropped from the process.

### 2.4. Reconstruction

All of the reconstruction process is implemented using the Open3D library [26], as it can process 3D data, and contains alignment algorithms, surface reconstructions, filters, and an integrated visualizer. The process starts by importing the five PCs and applying the transformation obtained from the extrinsic parameters. This transformation corresponds to the initial global alignment obtained from the extrinsic parameters during the calibration phase. Each PC suffers a rigid transformation by the 4 × 4 homogeneous transformation matrix.

Then, PCs are down-sampled by a scale of 0.002 for a faster processing time, being 0 the original size of the PC. Afterward, a refine alignment is performed with a pairwise registration for each acquired projection by using the Iterative Closest Points (ICP) algorithm, through a point to plane estimation with the objective function found in Equation (Equation 23) as follows:(23)$$\begin{array}{c} E\left(T\right)=\sum_{(p,q)\in \kappa }{\left(\left(p-Tq\right)n_{p}\right)}^{2} \end{array}$$
where *E* is the result from the objective function, T is the transformation matrix, κ is the correspondence set of points (p,q) from a target PC P, and a reference PC Q, and $n_{p}$ is the normal of point *p*. The ICP algorithm was preferred, as demonstrated to have the second best performance for aligning 2D PCs in previous work and since there is already an initial global alignment. The ICP is best suited as a refine alignment [27]. The pairwise registration is performed in the same order in which the cameras are physically placed on the frame. Once all of them are aligned, a Poisson surface reconstruction is performed [28]. This algorithm solves an optimization problem to obtain a smooth surface. Then, the resulting surface needs to be closed to be converted into a volume using MeshFix [29].

These previously mentioned algorithms that integrate the reconstruction pipeline of the system are then validated by evaluating the segmentation and reconstruction phases using region and distance-based metrics to quantify the performance of the algorithms used. These metrics are further described in the next section.

### 2.5. Validation

Validation is split into 4 blocks allowing different methods to demonstrate similarity, measurements, and soft tissue deformation capabilities. These are listed as follows:Segmentation comparison of a manual segmentation using image editing software (ground truth) vs. the use of the proposed algorithms.Comparison of a CT model (ground truth) vs. the scan with the proposed system.Diameter measurement verification with 4 plaster cast molds.Participant use demonstration with skin deformation.

For these comparison tests, Dice Coefficient (DC) and Hausdorff Distances (HD) are proposed to measure the error. On the one hand, DC calculates the similarity between two samples with Equation (Equation 24), where 1 is identical and 0 is no similarity.
(24)$$\begin{array}{c} \mathrm{dice}\left(A,B\right)=\frac{2|A\cap B|}{\left|A|+|B\right|} \end{array}$$
where A and B are binary images to be compared. The order does not matter, as the result would be the same.

On the other hand, the Hausdorff Distances (HD) metric measures the Euclidean distance between two sets with Equation (Equation 25) and returns the worst-case scenario, where 0 is the best and ≥0 is the worst outcome.
(25)$$\begin{array}{c} d_{HD}\left(A,B\right)=\max\left\{\underset{p\in A}{\sup}\underset{q\in B}{\inf}d\left(p,q\right),\underset{q\in B}{\sup}\underset{p\in A}{\inf}d\left(q,p\right)\right\} \end{array}$$
where A and B are binary images or PCs to be compared, “sup” and “inf” determine the least upper bound and the greatest lower bound, and *p* and *q* are points.

First, a total of four molds were scanned. The first mold corresponds to a 3D printed model to which its digital model was obtained from a CT image from the transtibial amputee participant M1 from Proactible, a local prosthetics clinic. The remaining three models correspond to plaster cast models obtained from Proactible. M2 belongs to a transtibial amputee, while M3 and M4 belong to transfemoral amputees. The segmentation results obtained in this phase were submitted to DC and HD metrics, obtaining a percentage value for DC and pixel count for HD, which can later be related using the system resolution obtained in Section 2.1.1.

Second, The M1 transtibial 3D printed model gets scanned with the proposed system. The 3D printed model was printed using HP’s Multi Jet Fusion technology with a ±0.3% (with a lower limit on ±0.2 mm) accuracy stated by the manufacturer [30] and this error would be expected for validation. Then, the reconstructed model obtained with the system is compared vs. ground truth via HD only. As the system outputs a surface mesh, HD is best suited for this application.

Third, the diameter of all four scanned models gets measured at the same view angle at three different heights. The bottom parts of the models were not considered to be included, as they serve as bases for the cutlines and are not important for the limb measurements.

Last, the system validates its use with a transtibial participant M6 from Proactible. This scenario shows the capabilities of the non-contact automatic reconstruction of the proposed system. Two scans at different poses were obtained during this phase. The first pose was obtained in a relaxed state, and the second by co-contracting the residuum. A comparison between these two poses was computed through HD to visualize muscle contraction with a distance heat map. This color map demonstrates skin deformation and geometries displacements.

The results are displayed and discussed in the following section, as well as the advantages of using this system and potential areas of improvement for future development.

## 3. Results & Discussion

This section presents the results to demonstrate the objectives previously presented. The proposed system accomplishes the mount of 5 RGB-D cameras and their calibration. It also can simultaneously capture images across all 5 cameras, segment, and reconstruct automatically a 3D surface mesh. Then, segmentation comparison using DC and HD takes place demonstrating a strong resemblance to the ground truth segmentation. Next, the 3D printed model obtained from a CT scan gets compared to the results of the proposed system using HD metrics. Then, all four scanned models with the system get compared by diameter measurement via both software and physically. Finally, the demonstration closes with the scan of a participant’s residuum.

### 3.1. ARACAM

Hereby, we present the prototype and test of a functional capture system. The system was used in two different configurations for the capture and digitization of shapes. The first configuration is used with plaster molds as shown in Figure 7, the structure is placed on a table covered by a green tablecloth. While the second configuration is used adjacently with participants as shown in Figure 8. Where it shows the workspace, the calibration support and the user interaction. The frame has a “C” shape cut out at the top so the participant can be moved closer to the frame. A 45° position of the frame was chosen so participants could easily place the residuum at the same orientation of the frame while sitting on a chair. Regardless of configurations, the frame demonstrates the camera mounting, the workspace, and the calibration support. This shows the frame holding 5 cameras with the ability to change orientation and position to direct the field of view to the target and calibration item. The frame can hold four cameras radially every 90° with 4 degrees of freedom (DOF) mount for each, by using a ball and socket joint combined with a rail axis as seen in Figure 8a, and a fifth fixed camera capturing the end view of the residuum. Moreover, all the cameras capture the desired object in place. An image of the workspace can be seen in Figure 8b displaying the setup of the frame before being used by a participant and operator. Looking at the top, the frame has a removable support to hold the calibration target as displayed in Figure 8c.

Wu et al. [14] reported the average processing time for the computer with their system to be around 19 seconds including calibration, scanning, registration, and reconstruction. Furthermore, they mention that they used 6 structured light cameras, which did not allow the system for simultaneous capture and make use of multiplexing the cameras on/off, an allowance that could increase the overall execution time for the system.

Simultaneous streaming was an issue in our previous work using structured light. With the use of stereoscopic cameras, the issue was solved in this work. Although the system is configured for the simultaneous stream, cameras can be synchronized via hardware to work at the same clock. This feature would enable synchronous video recording to record continuous tissue deformation.

### 3.2. Segmentation Results

Results for the segmentation process can be seen in Figure 9. This process shows the results for each step of the proposed algorithms, starting from the original image up to the segmented result. Since the binary image is the one that determines the end result of the segmentation. All binary images from the 4 models are compared via DC and HD metrics.

Figure 10 shows on the top part the ground truth segmentation, on the bottom the use of algorithms, at the top of each comparison displays the serial number for each camera. The footnote presents the numerical results corresponding to the DC and HD metrics. While on the one hand, DC presents a percentage, where digits closer to 1 means better, on the other hand, HD digits closer to 0 represent a better segmentation. Moreover, HD is measured in pixel count, error in mm may be obtained with the 0.24 $\frac{\mathrm{mm}}{\mathrm{px}}$ resolution previously calculated in Section 2.1.1. From the results presented in Figure 10 a summary of the statistics can be seen in Table 1 showing a mean of 0.87 for DC and 26.67 px or 6.40 mm for HD using the segmentation algorithms. This determines a strong resemblance to the ground truth segmentation. Other states are available in the table such as the upper (3rd Qu.) and lower (1st. Qu) quartiles, the Interquartile range (IQR) that shows the difference between the 3rd Qu. and 1st Qu., and the root mean squared (RMS) calculated from the DC and HD data itself. It is important to notice that different volume molds were used. For the segmentation phase, it can be seen in Figure 10 that models are arranged from smallest to largest. Since focal lengths and the cameras themselves are fixed smaller objects would occupy a smaller portion of the image resolution. For smaller objects, this could mean lower object resolution that could affect the error for the segmentation phase. In addition, there are errors due to the average and erosion filters that remove a portion of the edge. However, the areas of interest for reconstruction are the overlapping areas of the shape, not the edges. So these overestimations do not present a problem for reconstruction.

This results in a clean PC in Figure 11b, in contrast with a PC from the original scene as shown in Figure 11a.

For the validation of the results, it was sought to have a comparison of both regions and distance, so we proposed the use of DC and HD metrics. The Segmentation shows a strong visual correspondence between the ground truth and algorithms.

These tests show promising results, in this part of the segmentation we see errors due to the averaging and erosion filters that remove part of the edges. In this case, the sections of interest are those overlapping of the shape, not the edges. So these errors do not present a problem for reconstruction for this particular application. But the algorithms can be further improved. Currently, the algorithms employed to calculate the segmented region with the biggest element in the image make use of Otsu’s method to calculate areas within the image. Otsu’s algorithm may calculate wrong areas if lighting conditions are poor or over-exposed and/or depending on the image and targe colors. This specific algorithm works well with plaster cast models since they are white because brighter areas get marked in white and darker areas get marked in black. During the participant test, 2 out of 5 of the segmented images had trouble with the region selection due to this mentioned issue and manual intervention was needed. Since cameras can be automatically adjusted for light exposure, no additional light source was needed. But the use of an external light source could result in smoother and more uniform light along the target object while minimizing the shadows if the environment requires it.

### 3.3. 3D Reconstruction Performance Evaluation

The registration of the multiple PCs results in a clean 3D mesh in Figure 12b, in contrast with the 5 PCs global alignment shown in Figure 12a.

Next, Figure 13 presents the model comparison through HD for the M1 model. Where the original model in light blue was obtained with a CT scan, and the scan with the proposed system in yellow. A histogram of the HD results is shown at the left of the model, from the minimum value 0 mm up to the maximum value 10.7 mm going from blue through red respectively. Segmentation results for M1 can be seen in Figure 10a above. Then, Table 2 shows the statistical summary from the HD histogram presented, showing an error mean of 1.93 mm and a median of 1.27 mm.

The median is preferred over other statistics as it better describes the central tendency.

The bottom of the model colored in red represents the highest disparity for two reasons. First, the bottom part of the original CT model was modified for a proper 3D printing. Second, the disparity is due to the position of the model placed over the workspace. Adding to this, the system algorithms create a flat surface at the bottom where it touches the table. This region, although showing the greater error, is not important because it is below the cutline marked with magenta in Figure 13. This means that it is not part of the socket and, therefore, is not desired to be compared. The important region resides above the cutline showing a central tendency with a median of 1.27 mm of error. By comparison results with Wu et al’s foot scanner, report two accuracy validations by using a sphere with a real circumference of 942 mm and an average measured value of 940 mm for the scanned model. Additionally, they compared their system against a commercial laser foot scanner reporting the largest error of 5 mm and 80% of their data with ≤1.5 mm of error with a mean error of 1.42 mm [14]. Accuracy statistics from our previous work reported the use of a freeform and prismatic shape [9]. The scale in that study was updated to meet the same units in this work for a proper comparison. The freeform showed a max error of 8.3 mm with a mean of 1.3 mm and the tangram showed a max error of 4.8 mm with a mean of 1.3 mm. The characterization study for the SR300, demonstrated the multi-view performance using freeform and prismatic shapes with a max error of 2.5 mm [31]. Figures 17 and 18 in [31] show a lower and upper bound for their compared meshes, with no mean errors reported. The work of Solav et al. presented errors of 0.06 mm evaluating a cylinder with random patterns as the calibration target.

These mentioned studies related to the use of multi-view systems for reconstruction and the use of Euclidian distances for measuring error. The system presented here showed a median error of 1.27 mm which is less than the reported with some of the compared studies. The system proposed uses 5 stereoscopic cameras, while my previous work used 5 structured light cameras. Comparatively, Wu et al. used 6 structured light cameras and Solav et al. used 12 RGB cameras of 8 Mpx. The calibration procedure in this work utilizes a checkerboard using the Kabsch algorithm, previous work lacked a calibration target. Wu et al. use a T-shaped checkerboard pattern using a variation of Zhang’s method algorithm. Solav et al. use a cylinder with random dotted patterns. Another main difference regarding the calibration phase is that in this work and Wu et al. systems get calibrated previously during the setup stage, after that the checkerboards get removed before capturing phase. While Solav et al. paints a similar pattern over the subject that serves as markers to later correlate the images, resulting in less error reported from the discussed studies, but with the drawback of having to paint the test subject [15].

With the current system, the global alignment error can be reduced if the cameras setup is taken care of by obtaining values close to 0 during calibration with the Kabsch algorithm. In a practical way, we have noticed that placing the camera as close to being perpendicular to the board gets better results. However, the system also makes use of the ICP algorithm that depends on a threshold, which influences the final error of the model. So there is still room for improvement with the ICP algorithm to minimize this error.

These results allow us to discuss the ideal tolerances that should exist between the socket and the residuum. Works in the literature focus their efforts mainly on the reconstruction and manufacture tolerances, but no interface tolerances have been reported to date. This could be caused due to the skin’s elastic properties and ability to adapt to unconventional socket shapes. Traditional methods work on a trial-and-error basis in adding and removing material from the mold for socket manufacturing. Recent techniques with the use of medical images are very accurate and can have resolutions of 0.3 mm [8]. However, the use of FE models that optimize the shape by deforming the 3D model will depend on each participant, which will have variations for each socket. This would result in different tolerances for every socket [5]. This is still an area of opportunity that needs further research. Current methods try to approach these tolerances by using pressure sensors that can evaluate the interface pressure to understand the loading conditions and possible pain thresholds [32,33,34,35,36,37]. The resulting mesh from the proposed capture system can be further processed to meet desired diameters which would be well suited for socket tolerances and manufacture.

### 3.4. Plaster Mold Diameters Measurements

Furthermore, the scanned models get compared with the real diameter dimensions of the plaster cast models at the same view angle presented. Figure 14 demonstrates all four models with their corresponding measurements. It shows the diameters at three different heights obtained via software (Solidworks) of the scanned model and the physical counterpart. Additionally, M1 presents both the model obtained from the CT and the scanned model. The heights are proportionally distributed with respect to each model, meaning that the distances used vary and are not the same. This helps to verify that the measurements obtained from the scanned model resemble the physical model. Moreover, the different heights selected are considered to be below the cutline in the case of M1, M2, and M3. For M2, heights were considered to be below the knee, considering that the models are upside down. Additionally, the molds were arranged by size from small to large to demonstrate the dimension scan capacity of the system. Demonstrating that the system allows the scanning of small and large objects by adjusting the cameras with their respective 4 DOF. The largest mold tested has a diameter of ≈150 mm, as calculated in Equation (Equation 6) to ensure a proper field of view of the target.

The obtained measurements of the diameters enable a fast verification of measurements using a caliper, by taking measuring the diameter at 3 different heights demonstrates that overall, objects follow the same dimensions. As mentioned above in the study of Wu et al. they compared a real sphere vs. the scanned one having a difference of 2 mm for the circumference. The use of diameters from a single viewpoint was proposed due to the irregular shape of the amputation models which cannot be considered completely cylindrical. Initially, the measurement of the circumference was proposed but software limitations could not return the circumference, making the measurement of diameters at specific heights more viable. Overall error measurements remain ≤1.6 mm, having a strong resemblance with the real model. This proves to be useful for the assistance of the prosthetist in the prostheses manufacture. Facilitating the comparison of measurements of physical and digital models.

### 3.5. Participant Validation

Figure 15 shows the process from start to finish with the system previously calibrated. In the time span of 5 min, a full reconstruction from start to finish was accomplished with little intervention from the system’s operator. This time-lapse includes setting up the parameters of the camera and creating a folder for the current workspace. Furthermore, it includes executing the parts of the program that consider the calibration, image capture, segmentation, PC conversion, and reconstruction. The system’s operator intervention includes only inputting the data for the workspace creation, the placement, and removal of the calibration target, saving the desired images in the capture phase, and selecting the camera order in which reconstruction takes place. These may be performed previously as the setup stage, except the capture phase. Once concluded, the system is ready instantaneously for a full automatic reconstruction.

The 3D reconstruction results in Figure 16 show the comparison between two different poses. The first one is in a relaxed state and the second one performs muscle co-contraction. The quality map shows the differences and changes from one state to another showing great differences in 3 main parts, i.e., the knee, vastus lateralis (the external part of the quadriceps), and the residuum. The knee and residuum differences relate to the change in position, while the quadriceps region was caused by muscle contraction. Furthermore, an artifact appears at the same spot in both poses at the top part, making a sink. This could possibly be caused by the reconstruction algorithm while stitching the mesh.

Results show that the proposed system serves as a useful tool to scan and create a digital library for plaster cast models. Additionally, the diameters measurements verification demonstrates the use scenario to assist the prosthetist for residuum dimensions and manufacturing process. The 3D reconstructions of the transtibial and transfemoral plaster casts, as well as the reconstruction of the participant’s lower residual limb show that the ARACAM system is versatile due to the proposed design and the DOF of the mounted cameras. This allows acquisitions to be taken in different positions with minimal intervention or reconfiguration of the system. Looking back to the traditional pipeline in Figure 1, the integration of this system would be a hallmark for any prosthetics clinic to be able to create dossiers and see patients’ progress. The system would mean an update in the pipeline with the integration of the new technology as shown in Figure 17. This integration means the scan of the current plaster model can be re-fabricated and re-used in designing the definitive socket after 6–12 months have passed. Another possible change in future pipelines could be the replacement of the plaster cast models and directly scanning the amputee, resulting in a reduction of using materials and faster model iterations.

Additionally, throughout the literature review, there is already a new socket design paradigm using CT scans, which enables the location of external and internal geometries that can be used along with an FE-model for an automatic socket design [5]. The proposed system can be used to update the external geometries of previous CT models to avoid additional radiation exposure. More than that, future work will involve the use of the proposed system along with multi-view DIC to properly extract the viscoelastic tissue properties to feed an FE model.

Regarding areas for improvement, the reconstruction algorithm needs to input the camera order in which the reconstruction has to be carried out. This process requires the affixing of the first PC and making a pair-wise registration of the subsequent PC and so on until all PCs have had the ICP algorithm applied. While this registration works adequately, it is not ideal. Algorithms can be further improved so that key features can be detected and correspondences between images can be achieved to automatically determine the order in which registration should take place. Moreover, the use of body markers could be implemented to facilitate the detection of correspondences. This could also involve the calibration and the analyzing method used, which enables the possibility to implement a multi-view DIC to detect corresponding points across images.

## 4. Conclusions

The work presented addresses the design, prototype, and testing of a new fully automatic capture system employed specifically for lower-limb prosthetic design and manufacture. The prototype was designed and manufactured via laser cutting and bending of aluminum sheets that can support the mounting of 5 stereoscopic cameras, cameras with 3 DOF for positioning and orientation towards the target. The frame has a “C” shape cut-out for easy interaction with the participant. The system can perform an automatic 360° reconstruction including the end view of a target in less than 5 min. The system presents the advantage of simultaneously capturing across all RGB-D cameras. Added to this, it is calibrated to a global coordinate system via a checkerboard by implementing the Kabsch algorithm. Then, the system segments the acquired images to keep the target object, which is then mapped out into PCs with an initial alignment. Afterward, the registration algorithms align and reconstruct the surface from these PCs, resulting in a closed mesh.

The results for the plaster cast models show the segmentation and registration validations. The segmentation metrics of DC and HD show strong visual similarity. Moving forward, the 5-camera system was tested using a known prototype model obtained from a CT scan, to which comparison results via HD show an average error of ≤1.93 mm making the system competitive against the state-of-the-art reviewed systems. The different diameters obtained of the four models through SW present a reasonably close resemblance to the physical models. These results from the objectives show that the system is useful for the generation of a digital library to create a patient history to see the shape progression over time, as well as assist the prosthetist in the manufacturing process, the saved model can serve to update external geometries of previous CT scans, and future extraction of viscoelastic properties of the skin.

A full reconstruction was possible with only 5 images since no occlusion areas were observed with the convex shape of the amputee residuum including the molds, having no issue during the surface reconstruction with this application. Currently, registration still takes place by selecting the order in which cameras are placed. While results were promising, problems may arise if selecting the wrong camera order. Image correlation, key features, or even color ICP may be implemented to overcome this issue. Additionally, calibration via a traditional checkerboard method is sensitive to external disturbances and easily loses calibration, adding error to the initial global alignment. A cylindrical calibration target may be implemented along with the use of body markers to resolve this problem. The results obtained from the participant demonstrate that the system is capable of non-contact reconstruction and analysis. Cameras can be synchronized via hardware to ensure the same time frames across devices, which would enable the possibility of synchronous video recording to obtain an over-time shape deformation useful for the extraction of viscoelastic properties of the skin.

This work accomplishes to design, prototype, and test a new fully automatic capture system to scan plaster molds and serve as a platform for future technologies in lower-limb reconstruction applications. The challenges involved the selection and integration of technologies that could be useful to comply with the application of developing an automatic capture system. In less than 5 min the reconstruction of an object can be acquired with little operator intervention with the proposed system. Current commercial and industrial solutions for 3D reconstruction consist of large equipment for which a dedicated area is required for its operation, such as full-body scanning systems. This work addresses this issue by scaling down a multi-view system design to scan small objects and designed to consider involuntary or voluntary movements of the body or object during capture.

## Figures and Tables

**Figure 1 sensors-22-02443-f001:**
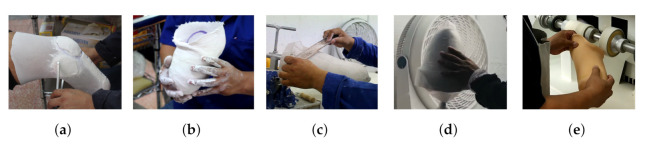
The overall process of conventional socket fabrication is shown. (**a**) Starting with the limb measurements. (**b**) Then, a plaster cast is obtained with the residuum geometry. (**c**) Later, the plaster cast model is post-processed by adding and removing material at key locations. (**d**) Next, lamination and curing take place. (**e**) Finally, the socket gets polished and the cut line is shaped.

**Figure 2 sensors-22-02443-f002:**
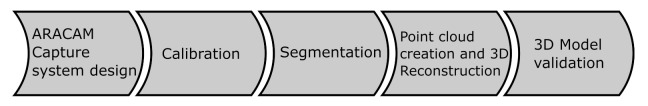
System pipeline.

**Figure 3 sensors-22-02443-f003:**
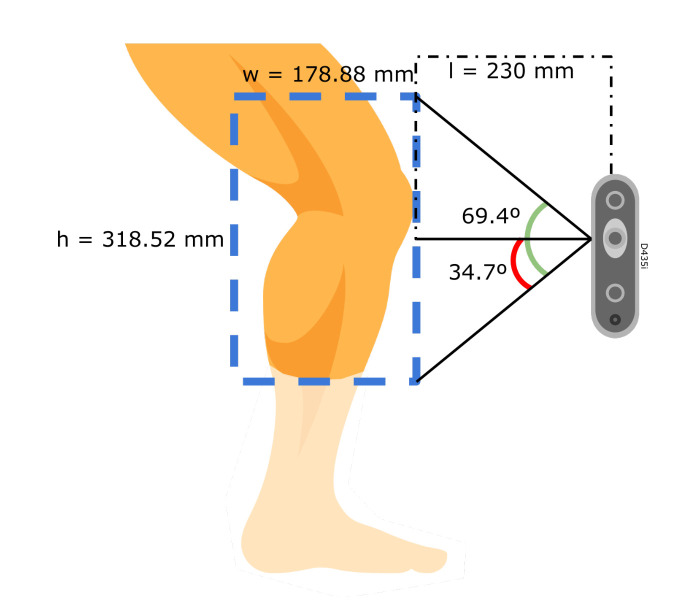
A vision area of 318.52 × 178.88 px at 230 mm from the target using the D435i stereo camera. This results in a resolution of 0.24 $\frac{\mathrm{mm}}{\mathrm{px}}$.

**Figure 4 sensors-22-02443-f004:**
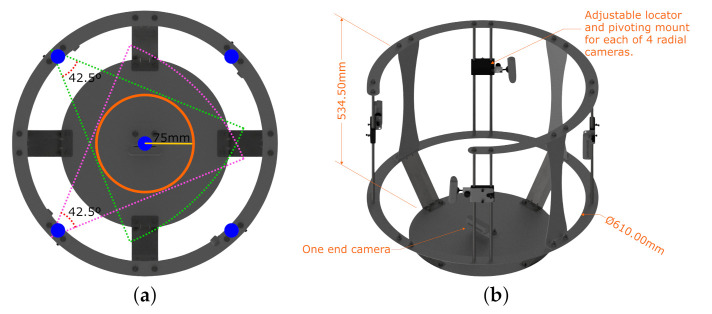
Mechanical design overview showing: (**a**) frame and camera setup with overlapping areas; (**b**) frame’s overall dimensions.

**Figure 5 sensors-22-02443-f005:**
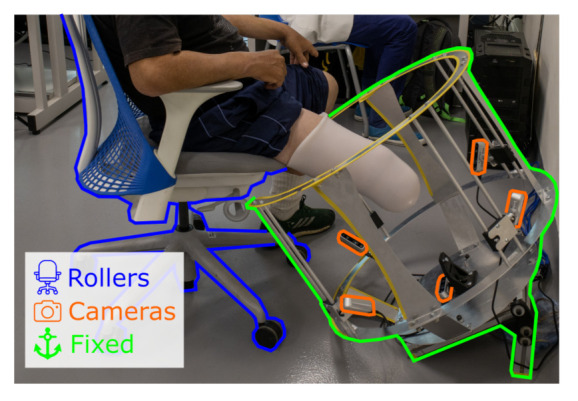
Example of the participant’s and operator’s interaction with the system, with a non-contact scheme. Labeled in blue, the rolling chair enables the participant to move towards the frame. In orange, previously calibrated cameras aiming at the residual limb. In green, the fixed structure, with a *C* shape at the top for ease of access.

**Figure 6 sensors-22-02443-f006:**
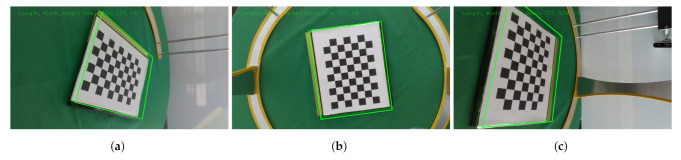
Calibration setup and the view from all 5 cameras (**a**–**e**). All sensors look at a common checkerboard serving as the calibration item. The green box displays the dimensions of the book underneath the checkerboard to validate that the process was completed successfully. This shows the simultaneous view of the calibration phase.

**Figure 7 sensors-22-02443-f007:**
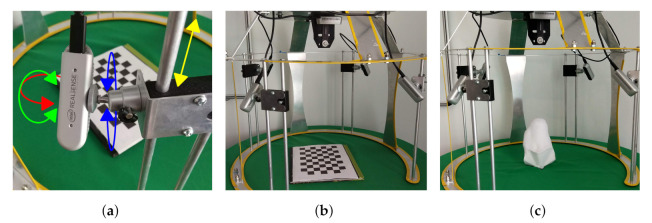
The configuration used for the plaster molds showing: (**a**) 4 cameras mounted on a support that enables rotation and positioning; (**b**) the workspace of the system being calibrated; and (**c**) the model being captured. Demonstrating the positions of camera mounts and their ability to change orientation to focus on the objective.

**Figure 8 sensors-22-02443-f008:**
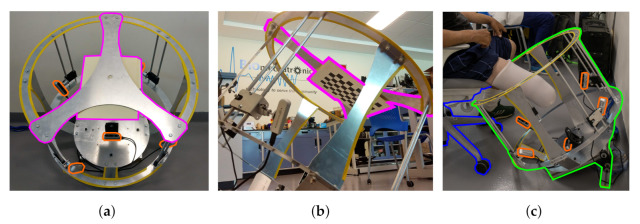
The configuration used for participants shows: (**a**) the workspace of the system being calibrated; (**b**) the removable support for target calibration in magenta; and (**c**) the interaction with the participant. Demonstrating the positions of camera mounts and their ability to change orientation to focus on the objective.

**Figure 9 sensors-22-02443-f009:**
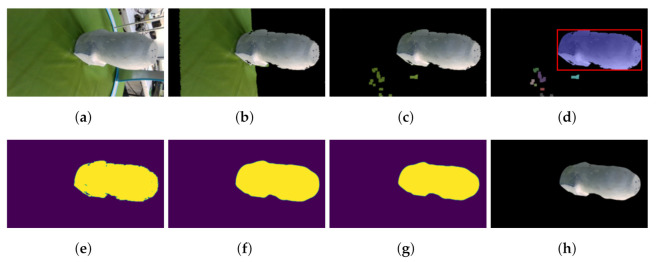
Automatic segmentation, feature extraction, contour smoothing, contour outlier removal, and masking process via the algorithms for one camera image. This shows the automatic segmentation from start to finish. (**a**) Original. (**b**) No background. (**c**) Color removal. (**d**) Region box. (**e**) Binary image. (**f**) Median. (**g**) Erosion. (**h**) Result.

**Figure 10 sensors-22-02443-f010:**
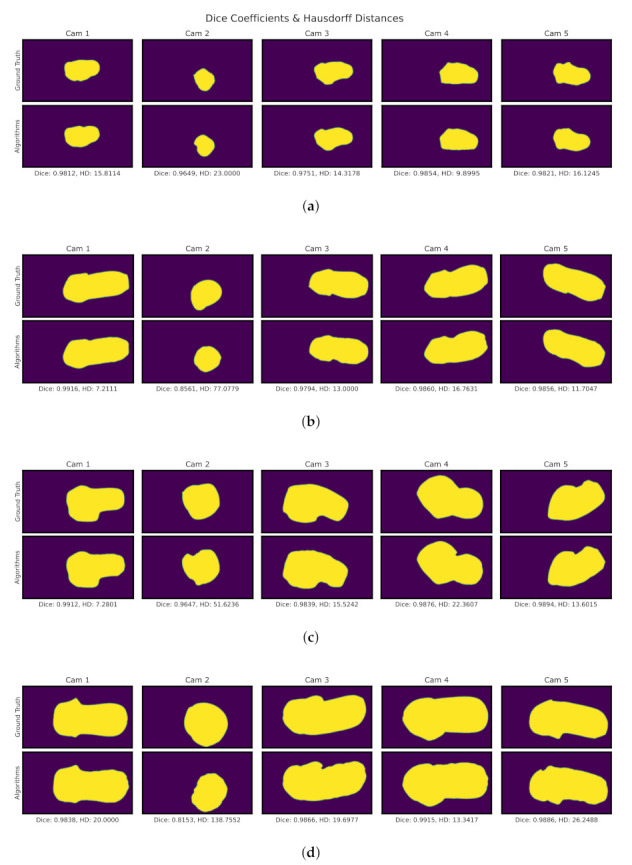
DC and HD metrics for RGB-based binary image segmentation for four scanned models. In each model, the top row displays the segmented object using manual segmentation as ground truth. Below are shown the segmentation results using the algorithms, which demonstrates a strong visual comparison. (**a**) M1. (**b**) M2. (**c**) M3. (**d**) M4.

**Figure 11 sensors-22-02443-f011:**
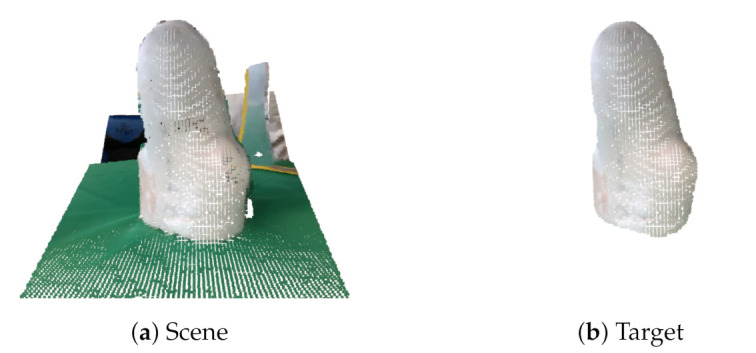
Comparison of the original scene (**a**) and the segmented target object (**b**). This demonstrates that the segmentation algorithms and PC conversion were working correctly.

**Figure 12 sensors-22-02443-f012:**
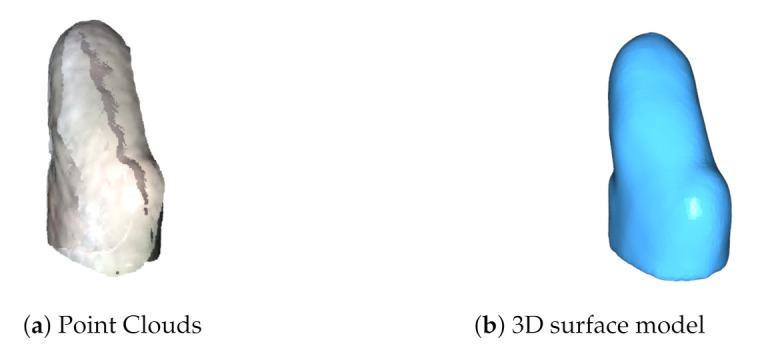
Comparison of (**a**) the initial global 5 images alignment and (**b**) the output model after applying a refined ICP registration and Poisson surface reconstruction. This shows the input and the output of the reconstruction algorithms.

**Figure 13 sensors-22-02443-f013:**
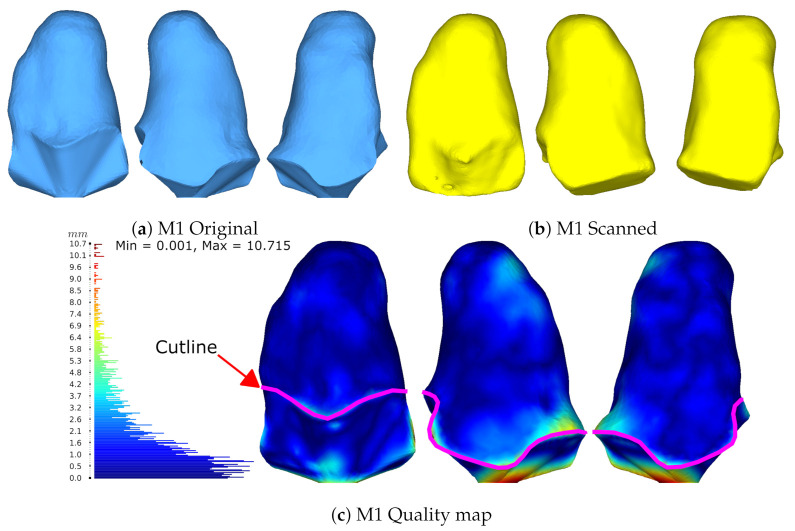
M1 model measured via HD. (**a**) The original model from the CT, (**b**) the scanned model with the proposed system, and (**c**) the quality map between the scanned and original models.

**Figure 14 sensors-22-02443-f014:**
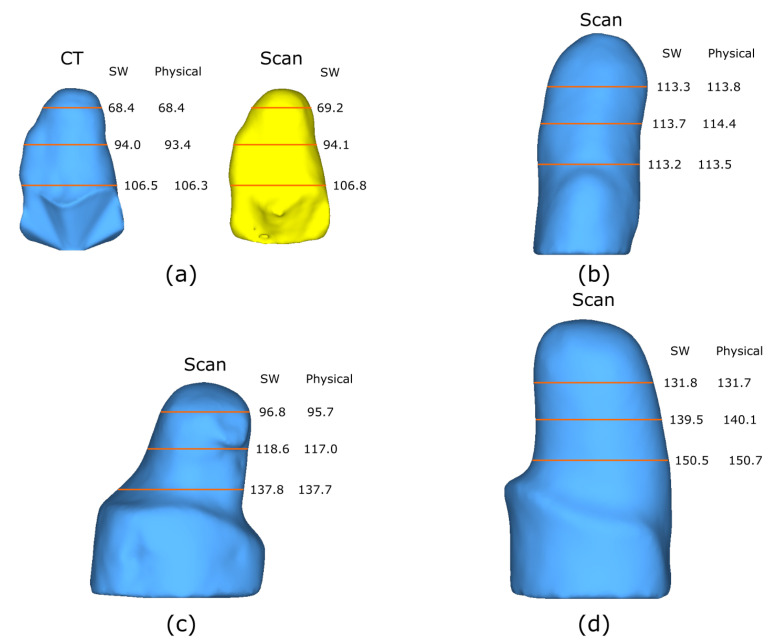
Diameter measurements across all 4 models. Comparison for both SW and physical diameters at 3 different heights (units are in mm). (**a**) M1-transtibial proto model. (**b**) M2-transtibial plaster. (**c**) M3-transfemoral plaster. (**d**) M4-transfemoral plaster.

**Figure 15 sensors-22-02443-f015:**
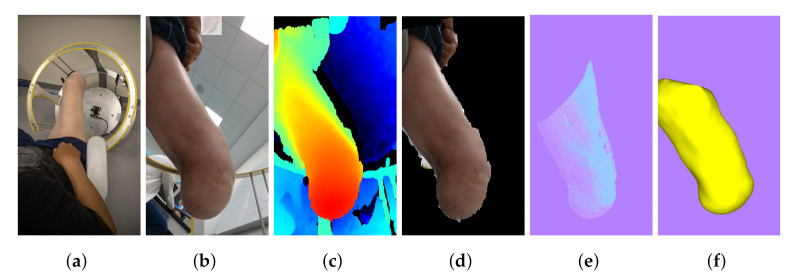
Testing of the system with a participant, starting from (**a**) the general view of the system and the participant, (**b**) the color image from one of the cameras, (**c**) the aligned depth image, (**d**) the segment of the target, (**e**) the mapping of the PC and (**f**) the resulting surface mesh after the 5 images reconstruction. This shows the process from start to finish of the system viewed from one camera angle.

**Figure 16 sensors-22-02443-f016:**
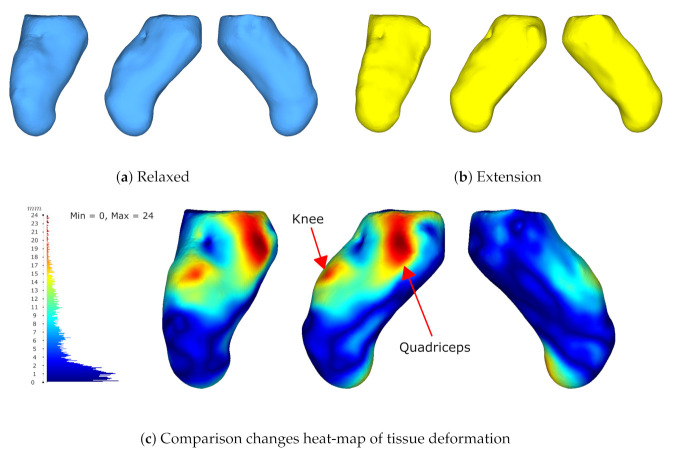
Results from two poses obtained from participant M6. Results measured via HD. (**a**) The residuum model in a relaxed pose, (**b**) the residuum model while co-contracting the muscles, and (**c**) quality map between both poses.

**Figure 17 sensors-22-02443-f017:**
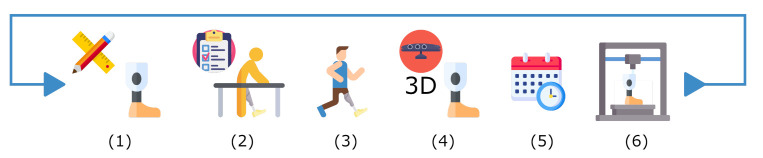
The traditional pipeline with the addition of technology integration. (1) Limb measurements and socket fabrication. (2) Check socket. (3) Provisional/Definitive socket. (4) Scan of current plaster cast mold. (5) 6–12 months before receiving a definitive socket. (6) 3D Re-fabrication of the mold from the digital library.

**Table 1 sensors-22-02443-t001:** DC and HD statistics obtained from the segmentation results.

Type	Min.	1st Qu.	Median	Mean	3rd Qu.	Max.	IQR	RMS
DC (%)	0.47	0.80	0.98	0.87	0.99	0.99	0.19	0.89
HD (px)	7.21	13.17	15.97	26.67	22.68	138.76	9.51	40.30
HD (mm)	1.73	3.16	3.83	6.40	5.44	33.30	2.28	9.67

**Table 2 sensors-22-02443-t002:** HD statistics results were obtained from the reconstruction model comparison (values in mm).

Min.	1st Qu.	Median	Mean	3rd Qu.	Max.	IQR	RMS
0.00	0.59	1.27	1.93	2.64	10.72	2.06	2.73

## Data Availability

The ARACAM code is available at https://github.com/marcoagbarreto/ARACAM (accessed on 1 March 2022).

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
