# Peer review of "ARACAM: A RGB-D Multi-View Photogrammetry System for Lower Limb 3D Reconstruction Applications"

_sensors, 2022, doi:10.3390/s22072443_

Round 1
Reviewer 1 Report
This is an important study to match the growth need for lower limb prostheses. The authors have reported detail experimental information with comprehensive scientific evidence to support their claims. The proposed research is well organized with decent investigation and evaluation, and will of great interests to the scientists in related areas.
In addition, this article is well written in English. Therefore, I would recommend it to be accepted as a hot article with special highlight.
Reviewer 2 Report
Dear Author, Dear Editors,
The article entitled “ARACAM: A RGB-D Multi-View Photogrammetry System for Lower Limb 3D Reconstruction Applications”, submitted for publication in Sensors, investigates the generation of lower limb prosthetic sockets through multi-view photogrammetry. Authors described in the introduction the clinical and technological issues induced by the current workflow to manufacture lower limb sockets, with a specific focus on the generation of a 3D model of the limb to adapt a patient specific socket. State of the art section is emphasizing the current trends to generate such virtual models and to automatize such process. The authors’ investigations using photogrammetric technology is then presented.
While interesting, this article needs to be thoughtfully reorganize to improve its clarity and the message the authors want to deliver. State of the art is quite long, could be far more concise and precise. The “contribution to the field” section could have been shortened and integrated as a conclusion of the state of the art if well written: it is still hard to understand what is the added value of this project. Technical sections seems to be quite solid, with small inaccuracies or abuse of language (cf. remarks). The main critic that could emerge is the non-physiological stance of the patient’s limb during acquisition (and so a high risk of mismatch between patient’s morphological features and the designed socket).
Specific Remarks:
- Something is missing line 32 “While several of the parts can be obtained in large quantities, materials, and sizes.”
- State of the art should be re-written focusing on the advantages and drawbacks of the different imaging modalities, to make the message clear: why choosing photogrammetry vs other technologies. What about 3D portative scanner that are cheap, mobile, and can give amazing results without such an expensive setting of yours?
- Contribution part should be removed and integrated as a conclusion of a compelling state of the art section
- The first paragraph of Material and Method section does not bring anything. Longer does not mean better. Often paragraph could be shortened to be more impactful.
- English should be edited, many adverbs (e.g., also), sometimes sentences are quite unclear. “Also, the cameras used allow simultaneous streaming across all 5 cameras”
- Please remove all the “in vivo” term you are using, it has nothing to do here.
- A few number of Figures or tables are missing (e.g., lines 500 & 649)
Reviewer 3 Report
I would like to congratulate authors for their work. The methodology shows a detailed description and its application can be considered an evolution for prothesis designs.
Different text style recommendations can be considered:
- Line 30. I recommend don`t use important two times
- The Figure description stile. Check the template.
- Line 235 “Next, The distance from…” The, use a capital letter.
- Line 252 remove the dot after “maximum”
- Line 255 and 256. After a dot use a capital letter.
- Line 262. Fig. 4a call has a double white space.
- Figure 6 size its very small. I recommend to increase the size.
- Line 478 require a reference.
- I recommend to increase the figure 9 size
- Line 500. Table ??
- Line 524. I recommend a reference for Otsu’s algorithm
- Line 543 3D in capital letter.
- Line 649. Figure ??
